# A Reliability Generalization Meta-Analysis of the Antisocial Process Screening Device

**DOI:** 10.3390/bs15070860

**Published:** 2025-06-25

**Authors:** Meng-Cheng Wang, Jiaxin Deng, Xintong Zhang, Jinghui Liang, Yiyun Shou

**Affiliations:** 1Department of Psychology, Guangzhou University, Guangzhou 510006, China; 2Department of Psychology, University of Bath, Bath BA2 7AY, UK; 3Department of Applied Social Sciences, The Hong Kong Polytechnic University, Hong Kong 999077, China; 4School of Psychology & Neuroscience, University of Glasgow, Glasgow G12 8QB, UK; 5Saw Swee Hock School of Public Health, National University of Singapore and National University Health System, Singapore 117549, Singapore; 6Lloyd’s Register Foundation Institute for the Public Understanding of Risk, National University of Singapore, Singapore 117602, Singapore; 7School of Medicine and Psychology, Australian National University, Canberra 2600, Australia

**Keywords:** meta-analysis, reliability generalization, antisocial process screening device, juvenile psychopathy

## Abstract

Although the Antisocial Process Screening Device (APSD) is one of the most widely used instruments for assessing psychopathic traits, the reliability estimates of the APSD show great heterogeneity across different studies. This investigation evaluated the reliability of the APSD using a reliability generalization meta-analytic technique across 158 studies (*N* = 75,749). The APSD demonstrated marginal to acceptable coefficient alphas ranging from 0.62 (for the Callous–Unemotional subscale) to 0.79 (for the total scale). Further moderation analysis revealed that the differences in administration formats significantly explained the variance of coefficient alphas for the APSD total and subscales, and the self-report version of the APSD manifested poorer coefficient alphas than other-report versions. The standard deviation of scale scores also partly accounted for the variance of the coefficient alphas. Overall, the APSD was found to be a reliable, practical measurement of psychopathic traits in youth, which can be widely applied in various study settings. Nevertheless, we recommend that parent- and teacher-report versions of the APSD as the preferred administering format of the measure when used for children and juveniles, while the self-report version of the APSD is recommended with caution when assessing youth psychopathy, unless multiple-assess methods are used.

## 1. Introduction

Psychopathic traits have features including interpersonal antagonism (e.g., superficial charm, egocentricity), distinctive affective dysfunctions (e.g., low empathy, remorselessness), and behavioral disinhibition (e.g., impulsivity, poor planning; [44]; [65]). Over the last decades, psychopathy as a concept has been extended to include children and adolescents ([30]; [60]; [75]; [76]). Consistent empirical evidence has shown that psychopathy in youth is closely correlated with aggression, disruptive behavior disorders, delinquency (e.g., fraud, theft), and greater criminal recidivism (e.g., [6]; [53]; [62]; [64]; [67]; [69]), and can predict adult psychopathy ([60]). Precise assessment of children and adolescent psychopathy has important theoretical and clinical implications on understanding the relationship between psychopathic traits and conduct problems ([12]; [74]), investigating clinical and predictive utility of externalizing behaviors ([75]), and guiding diagnostic classifications of childhood disruptive behavior disorders and developing targeted interventions ([31]). Several instruments were designed for use in children and adolescents, such as the Antisocial Process Screening Device (APSD; [33]), Youth Psychopathic Traits Inventory—Child Version (YPI-CV; [87]), and the Inventory of Callous–Unemotional Traits (ICU; [29]) which specifically captures the affective dimension of psychopathy. Among these instruments, the APSD was modeled directly after the Psychopathy Checklist-Revised (PCL-R; [43]). To date, the APSD ([33]; [35]) is one of the most widely used measures to assess psychopathic traits in children and adolescents. Nevertheless, reliability estimates of some of the APSD subscales are questionable (e.g., [22]; [67]; [93]). The present study focused on evaluating the overall reliability of the APSD scores across various test settings using Reliability Generalization (RG) analysis.

### 1.1. The Antisocial Process Screening Device (APSD)

The APSD is one of the most widely multi-informant measures for assessing psychopathy and antisocial behaviors in children aged 6 to 13 ([49]; [52]; [75]). To date, the APSD has been translated into various languages and used in a variety of countries ([25]; [71]; [72]; [79]; [93]). The items were adapted from the PCL-R ([43]) to be appropriate for children and youths. The APSD was initially developed as a scale rated by parents and teachers regarding target preadolescents ([35]). A self-report version was then proposed and favored in older children and adolescents ([56]; [61]), as the self-report scale’s reliability and validity were substantially greater for older children and adolescents than for younger children. The self-report scale is necessary when parents or teachers have little information about the target individual’s psychopathy, particularly in detained youths ([32]). The parent- and self-report ratings of the APSD have been shown to have moderate associations in previous studies of adolescents from different cultural contexts (e.g., Dutch, [21]; U.S., detained juveniles, [26]; Russian, [42]; U.S., non-referred children, [62]).

The subscales of the 20-item APSD vary in the existing literature due to inconsistent factor structures which have been reported across different samples. [35] ([35]) observed a three-factor structure comprising the original Callous-Emotional (CU; six items; e.g., “Good at keeping promises”, “Is concerned about the feelings of others”), Narcissism (NAR; seven items; e.g., “Uses or cons others”, “Blames others for mistakes”), and Impulsivity (IMP; five items; “Engages in risky or dangerous activities”, “Gets bored easily”) in both clinical and community samples in the United States ([32]). It has been argued that this three-factor structure is ideal for distinguishing the different components of psychopathy ([52]), and it has been supported by empirical evidence in both clinical and community samples (e.g., [6]; [28]; [40]; [53]; [79]). Thus, the current study focused on the three-factor model of the ASPD.

### 1.2. Internal Consistency Reliability of the APSD Scores

Reliability denotes the proportion of observed scores that are true scores ([3]; [8]). Reliability indicates the precision of the measurement and, in the case of the APSD scores, can influence the precision of assessing psychopathic traits ([2]; [41]).

Internal consistency is the most common form of reliability used in the literature ([45]), reflecting the degree to which test items scores capture a single latent construct coherently ([38]; [70]). Low internal consistency may suggest a significant proportion of measurement errors in or variability among the items. Cronbach’s alpha is one of the most commonly used measurements of internal consistency reliability reported for the APSD in previous studies.

The internal consistency reliability of APSD total and subscales have generally been in marginally to moderately acceptable levels. For example, [78] ([78]) obtained poor coefficient alphas for the APSD scores from a college student sample, ranging from 0.42 (for CU) to 0.78 (for total scale). In another study, the coefficient alphas for the subscales also ranged from poor to marginal levels (0.53 for IMP to 0.66 for NAR; [48]). However, the CU subscale tends to show the weakest reliability scores overall. For example, in one study, a coefficient alpha of only 0.26 was observed for the CU scores ([15]) while a large body of other studies reported coefficient alphas which were under acceptable levels (e.g., [36]; [92]). Some studies (e.g., [14]; [20]; [81]) have suggested that factors such as the small number of items in the subscales, a combination of positively and negatively worded items, inaccurate expression of items, and the affective, behavioral, and cognitive features of juveniles, may contribute to the low CU and IMP coefficient alphas.

Meanwhile, Cronbach’s alpha values for the APSD have varied across test settings and sample characteristics. Coefficient alphas of the APSD total score have been most homogenous for the teacher-report version (APSD-TR; ranging from 0.86 to 0.97; [4]; [25]), followed by the parent-report version (APSD-PR; from 0.73 to 0.89; [25]; [34]), and the other-report version (APSD-OR; 0.73 to 0.88, from psychiatrists or other caregivers; [21]; [63]). The coefficient alpha had the greatest variability for the self-report version (APSD-SR; from 0.62 to 0.88; [39]; [92]). In terms of sample characteristics, coefficient alphas of the total scale scores ranged from 0.73 to 0.91 when adolescent samples with a clinical diagnosis were the research targets of studies (e.g., [64]; [83]), from 0.70 to 0.97 when studies used community samples (e.g., [4]; [37]), from 0.68 to 0.88 when studies used detained samples (e.g., [63]; [92]), and from 0.74 to 0.88 when mixed samples were used (e.g., [27]; [48]). Likewise, sample size may also influence the coefficient alphas, for example, [21] ([21]) reported low coefficient alphas, particularly for the CU (0.54) and IMP (0.30) subscales in a small sample of 74 adolescents. Finally, the alphas of the APSD total scores might be influenced by scale language. Statistical support exists from intervals of alphas in both English (0.62 to 0.92; [34]; [92]) and non-English (e.g., Finnish version: 0.67 to 0.76, [54]; Portuguese version: 0.56 to 0.77; [66]; Persian version: 0.46 to 0.69; [24]) versions of the APSD.

The heterogeneity of the reliability estimates of the APSD may limit its predictive utility in research settings ([79]), especially considering the low internal consistency reliability. Therefore, a systematic investigation is needed to understand the potential factors that are likely to contribute to this poor reliability, and to provide evidence for whether the APSD is in fact an appropriate tool to be used in certain testing contexts.

### 1.3. The Current Study

The current study aimed to evaluate the internal consistency reliability (i.e., coefficient alphas) of the APSD and its subscale scores across different test settings using an RG meta-analysis. RG is a meta-analytic methodology which synthesizes reliability estimates across different studies ([85]). It offers a more compositive and steadier reliability estimate than any statistic derived from a single research sample. An RG study on the APSD helps assess the degree of variability in reliability coefficients across different types of measures, samples, and settings, and offers the opportunity to advance our understanding of factors that can affect reliability, thus allowing researchers and practitioners to be more informed when using the APSD in assessing juvenile psychopathy.

The current study had several intentions. First, we tested the average coefficient alpha for the APSD total and subscale scores. Based on previous empirical studies (e.g., [16]; [82]; [89]), we hypothesized that the total scores would exhibit an acceptable mean coefficient alpha while the CU subscale would have a relatively low mean coefficient alpha. Second, we examined the heterogeneity of the coefficient alphas across all studies, and expected to observe significant heterogeneity among them. Third, we investigated whether the heterogeneity of the coefficient alpha estimates could be explained by study or sample characteristics, such as administration formats, language of the measure, or sample age. As the self-report measures have already been shown to have advantages, such as more precisely detecting interpersonal and affective deficits ([57]), we expected that the APSD-SR would have higher coefficient alpha values than the other-report versions. Additionally, some research used translated versions of the APSD that had limited evidence for its translation quality (e.g., pilot studies or studies that tested the translation such as whether items were understandable and interpretable; [84]). In these cases, participants may not have correctly or precisely understood the translated APSD items (e.g., [6]; [25]). Therefore, it is reasonable to hypothesize that the alpha values of the APSD would be higher for the English version of the measure than for non-English versions.

For the sample characteristics, older adolescents generally have been shown to have a better understanding of the APSD items than younger individuals. Therefore, we expected that the alpha values would be lower for samples that had a lower mean age than for samples that had a higher mean age. Moreover, a positive relationship between the standard deviation of scores and the coefficient alphas was anticipated, on account of the classical test theory which contends that higher standard deviations of scores will cause larger coefficient alpha values when the standard error of measurement is constant ([17]). Finally, we explored the potential effects of other sample characteristics, including mean scores, gender composition, and standard deviation of participants.

## 2. Method

### 2.1. Literature Search and Selection

A literature search was performed using EBSCOhost, Elsevier, Google Scholar, PsycARTICLES, Psychology and Behavioral Sciences Collection, PsycINFO, and Springer, using the terms “Antisocial Process Screening Device”, “APSD”, and “Antisocial Process Screening Device and reliability”. The references of the retrieved studies were also checked. After discarding duplicates, studies were included based on the following three criteria: (a) studies were published in either Chinese or English; (b) they were empirical studies; and (c) they reported a sample-specific reliability estimate. Figure 1 shows the details of the literature selection process. A total of 158 studies which identified 361 coefficient alphas altogether were included in the final analysis. There were 97, 104, 82, and 78 coefficient alphas for the APSD total scale, CU, NAR, and IMP scales, respectively.

### 2.2. Data Extraction and Coding

To test how the potential study characteristics might have accounted for the variability of reliability, a range of information reported in the studies was extracted and coded. Coded variables at the categorical level were as follows: (a) scale version (i.e., APSD-PR, APSD-SR, APSD-TR, and APSD-OR); (b) country (i.e., United States vs. other countries); (c) scale language (i.e., English vs. non-English); and (d) sample type (i.e., clinical, community, detained, and mixed samples). The coded continuous variables were (e) mean age of target adolescent samples; (f) standard deviation of the age of target adolescent samples; (g) gender composition of the target sample (male%); and (h) sample size. Studies lacking information for these characteristics were labeled as being with missing values.[note 1] A predictive mean matching approach (PMM) was applied to replace missing data in continuous characteristics (i.e., characteristics [e] to [h]) with observations from complete cases which had similar characteristics but required further imputation. The PMM provides an unbiased estimate when sample sizes are small, or when data are seriously skewed ([50]). The variables that were used for matching these cases were from characteristics (e) to (h).

Along with these moderator variables, coefficient alpha values for the APSD total score and subscales were extracted based on three criteria. First, only the coefficient alphas of the 20-item APSD full scale and the 6-item CU, 7-item NAR, and 5-item IMP subscales were extracted to be included in the current analysis. Second, if the study was a longitudinal study, only the coefficient alphas collected at the baseline were extracted, given that the coefficient alpha would likely artificially increase due to the practice effect ([2]).

Third, when only one coefficient alpha was reported based on multiple samples, the sample characteristics of the multiple samples would be aggregated for that single alpha value. Appendix A presents the details of all the information extracted from the included studies.

The coding process was completed by four trained psychology students. To ensure the accuracy of the coded dataset, 20% of the observations and the corresponding information was selected at random and independently double-checked by two of the authors. Disagreements during the double-checking session were resolved through discussion and consensus.

### 2.3. Analytical Strategies

All analyses with untransformed coefficient alphas in the current study were carried out using the package *metafor* 2.4 ([90]) and *mice* 3.11 ([9]) in *R* program 3.3.3 for MacOS. Separate meta-analyses were managed for the reliability coefficients obtained from the APSD total scale and for each subscale of each measurement instrument. In all cases, random-effects models were used ([7]; [58]; [77]).

Heterogeneity was first estimated among the reliability coefficients in each meta-analysis using the *Q* statistic and *I*^2^ index, which were calculated as the quantification of the overall heterogeneity across studies. A *p* value for the *Q* statistic < 0.05 is an indicator of heterogeneity among coefficient alphas ([46]), while *I*^2^ values of approximately 25%, 50%, and 75% refer to low, moderate, and considerable heterogeneity across studies, respectively ([47]). To identify possible influential cases or outliers, the leave-one-out diagnoses were conducted. The leave-one-out diagnoses used studentized residuals, Cook’s distances, and changes in *Q* and *I*^2^. The studentized residuals and Cook’s distances were based on a mixed-effect linear meta-regression model without potential moderators. Potential influential cases were identified by inspecting the plots of the studentized residuals and Cook’s distances. We verified the influence of the identified cases by further examining the changes in *Q* and *I*^2^ for models for estimating mean coefficient alphas across scales and models for moderation analysis.

After examining heterogeneity existing across reliability coefficients, moderator analysis using mixed effect models was applied to quantify the degree to which the variability of coefficient alphas could be accounted for by study and sample characteristics. The effects of moderators were tested based on the Knapp and Hartung’s method ([51]; see also [58]; [91]). The statistical significance of the association was reflected by *p* values of *Q*_B_ and *t* statistics for categorical and continuous moderators. The *R*^2^ statistic estimates the proportion of variance which is accounted for by the moderator variables ([59]).

Finally, the publication bias, namely file-drawer problems, was examined by funnel plots and the trim-and-fill method ([23]). Funnel plots for the APSD total scale and subscales were constructed to allow the threat of missing possible effect sizes to be assessed graphically. Meanwhile, the trim-and-fill method provided a null hypothesis test which assumes the number of missing studies (which refers to coefficient alphas in the current study) is zero, while Begg’s rank test ([5]) examined whether the observed coefficient alphas correlated to the corresponding sampling errors. A high Kendall’s tau rank correlation coefficient computed by Begg’s rank test can indicate an asymmetric funnel plot, implying a potential publication bias.

## 3. Results

### 3.1. Description of the Data

The literature search identified 361 coefficient alphas from 158 studies, with a total sample size of 75,749. Most coefficient alphas were observed from scores of the APSD-SR total scale and subscales, with a total number (*k*) of 189 coefficient alphas (52.35%), followed by the scores of the APSD-PR (*k* = 87, 24.10%). Of the 361 coefficient alphas, only a small number of them were obtained from translated or adapted versions of the APSD (*k* = 84, 23.27%), and a total of 189 coefficient alphas (52.35%) were from studies conducted in the United States. Finally, 49.58% of the coefficient alphas (*k* = 179) were reported from community samples, while 24.38% (*k* = 88) were from detained samples, 11.36% were from mixed samples (*k* = 41), 13.30% were from clinical samples (*k* = 48), and 1.34% (*k* = 5) did not specify the sample type.

### 3.2. Mean Reliability and Heterogeneity

As shown in Table 1, the overall (weighted) coefficient alphas ranged from 0.62 (CU) to 0.79 (total scale). These alpha coefficients indicated a statistically significant heterogeneity across studies with significant *Q* values of between 848.965 and 4106.680 (all *p*s < 0.001), and *I*^2^ values exceeding 90%.

A detailed inspection of the data indicated two possible influential studies accounting for this considerable heterogeneity. First, the [4] ([4]) study was identified as an influential case for the APSD total scores. This research reported the highest coefficient alpha (0.97) among all studies observing coefficient alphas from total scores. This study was the only study that used the teacher/psychologist version of the APSD, while other other-report studies only used the teacher-report version. Results from the leave-one-out diagnoses marked this study as an influential case with a residual term (3.503) from the mixed-effect model, together with a large Cook’s distance (0.094), compared to other observations. Its removal led to reduced heterogeneity *Q* (Δ*Q* = 2683.501) and *I*^2^ (Δ*I*^2^% = 1.93) indices.

Another potential influential case was the [21] ([21]) study, which administrated the IMP subscale in a community adolescent sample. This study reported a rather low coefficient alpha of the IMP subscale (i.e., 0.30) and contributed to a great amount of heterogeneity. The leave-one-out diagnoses also indicated that this study could be an outlier and an influential case, reflected by a large residual term from the mixed-effect model (−3.686) and Cook’s distance (0.184). Excluding the [21] ([21]) study led to decreases in *Q* (Δ*Q* = 40.532), *I*^2^ (Δ*I*^2^% = 1.03), and tau^2^ (τ^2^ = 0.005 with its removal) values for the IMP (see Table 1).

However, both studies met our inclusion criteria and did not substantially influence the findings in our analyses, including the pooled effect sizes (i.e., mean coefficient alphas). Thus, both studies were retained in the analyses. Sensitivity analyses omitting Barrutieta and Prieto-Ursúa for total score and [21] ([21]) for IMP were conducted, and the results are presented in the Appendix A.

### 3.3. Moderator Analysis

Table 2 presents the estimated alpha values for APSD based on administration formats. We found that the APSD-TR had the highest mean coefficient alphas (0.90 for total scale, 0.73 for CU, 0.85 for NAR, and 0.77 for IMP), while the APSD-SR had the lowest mean coefficient alphas, ranging from 0.51 for CU to 0.77 for the APSD total scale. The administration format accounted for 43.90% (*k* = 97) of the total variance of the alpha value of the total scale (*p* < 0.001), 47.40% (*k* = 104) for the CU (*p* < 0.001), 56.70% (*k* = 82) for the NAR (*p* < 0.001), and 44.50% (*k* = 78) for the IMP scales (*p* < 0.001).

Studies conducted in the United States yielded similar mean coefficient alphas (i.e., 0.79 for total scale, 0.62 for CU, 0.65 for IMP, and 0.74 for NAR) to those conducted in other countries, such as Belgium, China, and Turkey (i.e., 0.78 for total scale, 0.61 for CU, 0.66 for IMP, and 0.73 for NAR). Likewise, there were no significant differences in the total APSD and subscale scores between the original English version and other translated versions (i.e., Chinese, Dutch, Portuguese versions).

As demonstrated in Table 3, only the sample type explained a statistically significant amount of variance in the reliability estimates for the CU (*p* = 0.044), accounting for 7.10% (*k* = 104). Although the effects were not statistically significant for the total, IMP, and NAR estimates, clinical samples generally yielded relatively higher mean coefficient alphas for total scale (i.e., 0.82), NAR (i.e., 0.77), and IMP (i.e., 0.67) than non-clinical samples. The detained samples showed relatively lower mean coefficient alphas for the total scale (i.e., 0.78), CU (i.e., 0.56), and NAR (i.e., 0.72), but higher for IMP (i.e., 0.64) compared with mixed samples. Regarding the continuous variables analyzed using meta-regression (see Table 4), the standard deviation of scores showed significant positive relationships with the coefficient alphas for all scales (estimates ranging from 0.023 for the total scale to 0.060 for IMP, all *p*s < 0.05), except for CU (*b* = 0.040, *p* = 0.058). Standard deviation accounted for 20.70%, 15.90%, and 32.50% of the variance of the coefficient alphas for the total scale, NAR, and IMP, respectively. The mean age of the adolescents had a significantly negative association with the coefficient alphas for the APSD subscales, with estimates ranging from −0.013 for IMP (*p* < 0.001) to −0.005 for the total score (*p* = 0.009), and accounting for 8.50% (total score), 8.40% (CU), 23.70% (NAR), and 28.70% (IMP) of the variance of the coefficient alphas. The standard deviation of the assessed participants’ age was found to have a weak but significant positive association with the coefficient alpha for IMP with an estimate of 0.024 (*p* = 0.044), accounting for 5.70% of the variance. None of the other continuous variables significantly explained the variance of the coefficient alphas.

### 3.4. Explanatory Models

Weighted multiple regression analysis was conducted assuming mixed-effect models for the purpose of identifying the best set of moderator sets which would explain the variance of the coefficient alphas for the APSD total and subscales.

For the APSD total, the explanatory model initially included the administration format, mean of adolescents’ age, and the standard deviation of the scores as three predictors. Although the full model was statically significant (*k* = 41, *F* = 3.993, *p* < 0.001), the estimates of mean age and standard deviation of scores were not significant (*p* = 0.798 for the mean age and *p* = 0.444 for the standard deviation of scores). Therefore, only the administration format was retained in the final explanatory model as indicated in the meta-ANOVA results.

As for CU, the administration format, sample type, and mean age of the target adolescent sample were included in the multiple meta-regression analysis. When the administration format was a part of the model, neither of the remaining two moderators or their combinations remained statistically significant. As such, the administration format was the only moderator for this explanatory model.

With regards to NAR, three moderators, including administration format, mean age of target adolescents, and standard deviation of subscale scores, were included in the model. In the subsequent analysis, the regression estimate of the age of target adolescents was not significant. The final model which incorporated the administration format and standard deviation of NAR scores was statistically significant (*k* = 52, *F* = 17.881, *p* < 0.001, accounting for 66.22% of the total variance).

For IMP, the administration format, mean age of target adolescents, standard deviation of adolescents’ age, and the standard deviation of observed scores were included in the initial explanatory model. Only the standard deviation of observed score was significant (*p* = 0.006). Hence, this is the only moderator retained in the explanatory model, with the results presented in Table 4.

Overall, the results of the model misspecification test in all explanatory models were statistically significant (*p*s < 0.001), which indicated that there were other possible moderators also affecting the coefficient alpha variabilities of the APSD total and subscale scores.

### 3.5. Publication Bias

Publication bias in our results was detected by constructing funnel plots, as exhibited in Figure 2, and then trim-and-fill analysis ([23]) was used to examine the symmetry of the funnel plots. Results demonstrated that no missing study was found to violate the symmetry of the funnel plots with regards to the total score, NAR, or IMP, with acceptances of the null hypotheses (all *p*s = ns). However, the trim-and-fill method indicated 13 coefficient alphas on the funnel plot for CU (indicated by white dots), along with the rejection of the null hypothesis that no missing coefficient alphas on the right side of the funnel plot would affect its symmetry (*p* < 0.001). This could be due to very small sample sizes or high standard errors ([80]). The result of Begg’s rank test for CU was also statistically significant (Kendall’s τ = −0.208, *p* = 0.002). Thus, publication bias may be present for the CU subscale.

## 4. Discussion

The current RG meta-analysis evaluated the coefficient alphas of APSD, and examined the factors that might contribute to the variability of its reliability coefficients. The internal consistency coefficients as indexed by Cronbach’s alpha of the APSD fluctuated across different application contexts.

We found that the APSD total scores had an adequate average coefficient alpha, while the CU and IMP subscale scores had less satisfactory coefficient alphas. The low alpha values of these two subscales could be due in part to the small number of items in these subscales ([86]; [14]) or insufficient coverage of the symptoms by the limited items.

For the CU subscale, two factors might contribute to the low alpha values. First, the CU subscale has a mixture of positively and negatively worded items, which may artificially create a multidimensionality structure of the scale, thus also contributing to the low coefficient alpha value ([81]). Second, it might be difficult for participants and observers to detect unemotional traits, which are mostly about an absence of emotions. Absence of emotions may also be mistaken for other factors such as anhedonia, or the subject being withdrawn or shy ([10]).

The low coefficient alpha of the IMP subscale, particularly for the self-report version, might be due to some of the symptoms measured by the subscale items being confounded by factors other than impulsiveness. For example, Item 1 (“Blames others for their mistakes”) has strong implications of interpersonal or social outcomes. This item could be influenced by impulsiveness, general risk-seeking in the social domain, or social boldness. On the other hand, items such as Item 13 (“Engages in risky and dangerous behavior”) and Item 17 (“Does not plan ahead”) place the emphasis on the occurrence of the behaviors rather than on the motivation of the individual. In addition, [20] ([20]) have suggested that adolescents might not accurately assess whether they are engaging in dangerous activities or not recognize potential risks due to their limited cognitive ability. Adolescent examinees may misread the test items or misjudge their behavioral engagement.

### 4.1. Moderators of the Coefficient Alphas

We found that differences in the administration formats of the measure significantly explained the heterogeneity in coefficient alphas in the APSD total as well as in its subscales. The self-report version of the APSD presented the lowest average Cronbach’s alpha. Self-reports have been assumed to outperform other versions only when subjects can and do accurately assess their symptoms and behaviors ([88]). However, youths may have difficulty identifying and reporting their personal feelings correctly ([11]) which may increase the measure error of the self-report versions of the ASPD. On the other hand, as the APSD items focus on adolescent behaviors, observers (e.g., parents, teachers) demonstrate more a reliable assessment of overt behavioral symptoms, while adolescent self-reports show limitations in evaluating their own symptoms due to their incapacity to self-monitor ([32]; [68]). These findings underscore the critical role of informant discrepancies in child psychopathology assessment, highlighted by the Attribution Bias Context Model (ABC Model) ([19]; [18]). For example, older children and adolescents typically demonstrate greater self-awareness of their psychological symptoms compared to younger children, leading to improved concordance between self- and observer-reports. Moreover, the magnitude of discrepancy may vary by symptom domain: externalizing problems generally show higher inter-rater agreement due to their observable nature, whereas internalizing problems exhibit greater discrepancies as they rely more heavily on subjective experience. These patterns highlight the need for future research utilizing comprehensive multi-informant, longitudinal designs to elucidate the nature of observed discrepancies in psychopathology assessment.

We also found that sample type had a significant moderation effect on the heterogeneity in the alpha values in the CU subscale. The detained samples had the lowest and most unacceptable mean coefficient alphas. This could be because detained youths often tend to display more fraudulent or defensive response styles to conceal their true thoughts or sentiments ([13]). Detained samples also tend to have a relatively low level of education ([55]), which limits their comprehension abilities when it comes to the CU items. Furthermore, some CU items might not be applicable to incarcerated juveniles. For example, Item 3 (“Concerned about schoolwork”) and Item 20 (“Keeps few of the same friends”) focus on interpersonal situations in schools and communities, which may be less applicable to detained samples. Finally, all alpha values for the detained samples were based on the self-report version CU subscale (see Appendix A), which could also be another reason for the low alpha values for detained samples. This was also reflected in the fact that the sample type effect diminished when the scale version was entered into the explanatory model.

As for the continuous variables, the standard deviation of the subscale scores significantly explained the variance of the coefficient alphas in both NAR and IMP subscales. This finding is consistent with the predictions of psychometric theories that the larger the standard deviation of the score, the larger the coefficient alpha ([17]). More specifically, the standard deviation is the square root of the variance, and the alpha is larger if the value of variance is larger according to the formula for the Cronbach’s alpha. Therefore, the standard deviation of the subscale may be a moderator for the variation in each scale’s coefficient alpha.

Furthermore, the mean age of the target adolescents was negatively associated with the coefficient alphas in all three APSD subscales. However, when included in the final explanatory model, this association became insignificant. Nevertheless, [36] ([36]) have noted an insignificant correlation between psychopathic traits and age within a large sample of East Asian schoolchildren. More research, particularly longitudinal studies, are needed to further understand this correlation.

### 4.2. Publication Bias

Publication bias was found in terms of the reported alpha values of the CU subscale, with an underreporting of high alpha values, specifically. In light of this, the mean coefficient alpha of the CU subscale should be regarded as an underestimated value. The possibility for obtaining a high coefficient alpha for the CU subscale was limited because of the reasons mentioned above, which include mixed item wording, the limited number of items, and the CU trait’s low construct stability. Overall, caution is needed when interpreting the results of CU alpha values, and more research into this subscale specifically is needed.

### 4.3. Recommendations, Limitations, and Future Directions

The average coefficient alphas for the APSD total and subscale scores provide a guide as to the assessment precision of the APSD across different test settings. The results of the explanatory models can also help researchers evaluate the factors that may affect the coefficient alphas of the APSD scores. According to our findings, the APSD has relatively acceptable internal consistency reliability, and both the total and subscale scores of the parent-report and teacher-report versions of the APSD exhibit good alpha reliability. We recommend that the informant-report version of the APSD be preferred when using the APSD among youths, and that multiple-assess methods should be considered ([1]; [64]; [73]).

Some limitations should be noted for the current study. We focused primarily on the studies that adopted the three-factor structure of the APSD. Future research could consider other forms of the APSD to test the generalizability of the current findings. Publication bias in the CU subscale’s alpha values also limits the generalization of the results of this study. Finally, the test of model misspecification indicated statistically significant residual heterogeneity, which demonstrated that moderators other than those included in the current study may also affect the variance of the coefficient alphas. Future studies should investigate a wider range of potential moderators.

## Figures and Tables

**Figure 1 behavsci-15-00860-f001:**
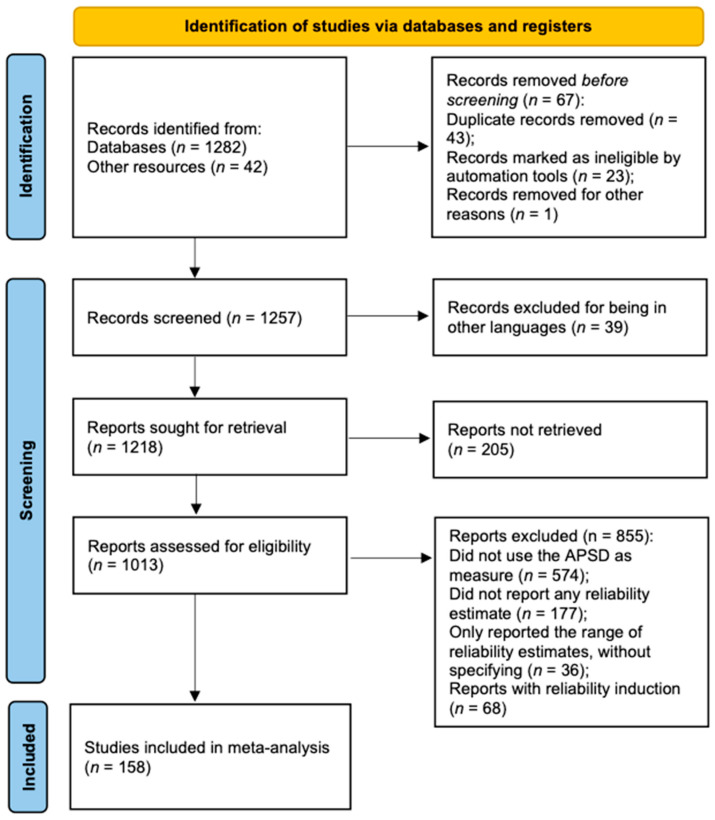
Flow diagram of study selection process.

**Figure 2 behavsci-15-00860-f002:**
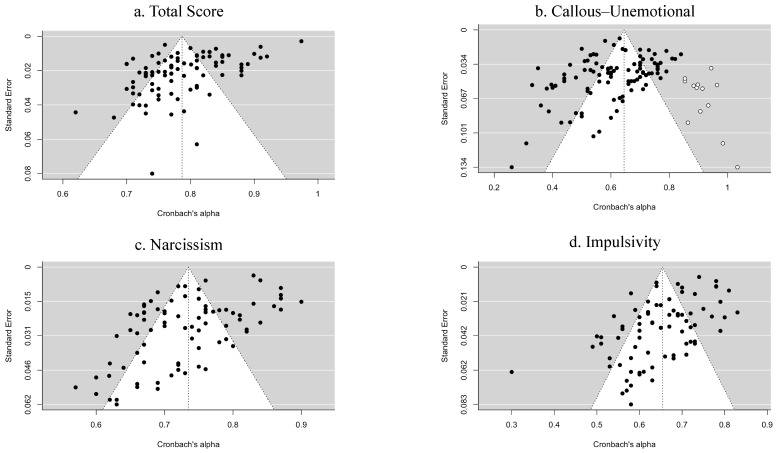
Funnel plots of total scale, callous–unemotional subscale, narcissism subscale, and impulsivity subscale. *Notes.* Black dots = observed studies; white dots = imputed missing studies estimated by the trim-and-fill method.

**Table 1 behavsci-15-00860-t001:** Mean effect sizes and heterogeneity statistics of the APSD.

	*k*	α*_+_*	95% CI	*Tau* ^2^	*Q*	*I*^2^%	*H* ^2^
Total Score	97	0.79	[0.77, 0.80]	0.003	4106.680 ***	94.65	18.68
Total Score ^a^	96	0.78	[0.77, 0.80]	0.003	1423.179 ***	92.72	13.74
Callous–Unemotional	104	0.62	[0.59, 0.64]	0.012	1007.952 ***	92.22	12.86
Narcissism	82	0.73	[0.72, 0.75]	0.004	1379.95 ***	94.16	17.12
Impulsivity	78	0.65	[0.63, 0.67]	0.006	848.965 ***	92.52	13.37
Impulsivity ^b^	77	0.66	[0.64, 0.68]	0.005	808.883 ***	91.52	11.79

*Notes. k* = Number of coefficient alphas; α_*+*_ = Mean coefficient alpha; 95% CI = 95% confidence interval for α_*+*_; *Q* = Heterogeneity statistic, *df* = (*k* − 1); *Tau*^2^ = Estimated total heterogeneity; *I*^2^ = Heterogeneity index. ^a^ = Results of impulsivity with [4] ([4]) study; ^b^ = Results of impulsivity without [21] ([21]) study. *** *p* < 0.001.

**Table 2 behavsci-15-00860-t002:** Comparison of coefficient alphas of the subscales across categorical variables.

	Total Scale			Callous–Unemotional			Narcissism			Impulsivity		
	*k*	α*_+_*	95% CI	*k*	α*_+_*	95% CI	*k*	α*_+_*	95% CI	*k*	α*_+_*	95% CI
Administration Format												
APSD-PR	15	0.81	[0.78, 0.84]	31	0.62	[0.58, 0.65]	20	0.75	[0.73, 0.78]	21	0.69	[0.66, 0.72]
APSD-SR	70	0.77	[0.76, 0.78]	33	0.51	[0.48, 0.55]	44	0.70	[0.69, 0.71]	42	0.62	[0.60, 0.64]
APSD-TR	5	0.90	[0.88, 0.92]	15	0.73	[0.69, 0.77]	9	0.85	[0.83, 0.87]	7	0.77	[0.74, 0.81]
APSD-OR	7	0.85	[0.79, 0.90]	25	0.68	[0.65, 0.71]	9	0.76	[0.70, 0.81]	8	0.65	[0.56, 0.74]
Country												
United States	43	0.79	[0.77, 0.81]	58	0	[0.59, 0.65]	44	0.74	[0.72, 0.76]	44	0.65	[0.63, 0.68]
Other	54	0.78	[0.77, 0.80]	46	0.61	[0.57, 0.65]	38	0.73	[0.71, 0.75]	34	0.66	[0.62, 0.69]
Scale Language												
English	50	0.79	[0.77, 0.81]	74	0.62	[0.59, 0.65]	58	0.74	[0.72, 0.76]	58	0.65	[0.63, 0.68]
Non-English	37	0.79	[0.77, 0.80]	21	0.59	[0.54, 0.65]	14	0.73	[0.69, 0.76]	12	0.66	[0.61, 0.70]
Sample Type												
Clinical Sample	6	0.82	[0.77, 0.87]	20	0.64	[0.58, 0.70]	10	0.77	[0.73, 0.82]	12	0.67	[0.63, 0.72]
Community Sample	45	0.78	[0.77, 0.80]	52	0.61	[0.57, 0.64]	42	0.73	[0.70, 0.75]	40	0.66	[0.63, 0.69]
Detained Sample	32	0.78	[0.75, 0.80]	15	0.56	[0.49, 0.63]	21	0.72	[0.69, 0.75]	20	0.64	[0.61, 0.67]
Mixed Sample	13	0.79	[0.77, 0.82]	17	0.67	[0.63, 0.71]	6	0.76	[0.71, 0.80]	5	0.62	[0.52, 0.71]

*Notes. k* = Number of coefficient alphas for each subgroup of the moderator variable; α*_+_* = Weighted mean coefficient alpha for each subgroup of the moderator variable; APSD-OR refers to the other-report versions of the APSD (i.e., staff-report, psychiatrist-report, and caregiver-report); Other countries refers to countries such as Australia, China, The Netherlands, etc. (see Appendix A); Non-English language refers to languages such as Chinese, Dutch, Portuguese, etc. (see Appendix A).

**Table 3 behavsci-15-00860-t003:** Meta-ANOVA by categorical moderators.

	Total Score			Callous–Unemotional			Narcissism			Impulsivity		
	*R* ^2^	*Q* _B_	*p*	*R* ^2^	*Q* _B_	*p*	*R* ^2^	*Q* _B_	*p*	*R* ^2^	*Q* _B_	*p*
Administration Format	0.439	60.412	**<0.001**	0.474	71.297	**<0.001**	0.567	76.635	**<0.001**	0.445	40.292	**<0.001**
Country	0.005	0.327	0.567	0.001	0.287	0.592	0.006	0.352	0.553	0.001	0.020	0.888
Language	0.000	0.001	0.979	0.01	1.080	0.299	0.009	0.275	0.600	0.000	0.005	0.946
Sample Type	0.026	2.483	0.478	0.071	8.090	**0.044**	0.059	5.161	0.160	0.032	2.047	0.563

*Notes. R*^2^ = Proportion of variance explained by the moderator variable; *Q*_B_ = Between-group heterogeneity test; *p* = *p* value for the statistical tests of *Q*_B_. Significant results are in bold.

**Table 4 behavsci-15-00860-t004:** Results of simple meta-regression analysis.

	*k*	*b*	*t*	*p*	*R* ^2^	*Q* _E_
Total Score						
Age Mean	87	−0.005	−2.614	**0.009**	0.085	1023.132 ***
Age *SD*	86	0.005	0.797	0.425	0.008	1155.398 ***
Scores Mean	47	−0.003	−1.692	0.091	0.065	567.180 ***
Scores *SD*	46	0.023	3.024	**0.002**	0.207	448.175 ***
Male % of sample	91	0.000	−0.049	0.961	0.000	3011.995 ***
Sample Size	97	0.000	−1.282	0.200	0.025	3462.726 ***
Callous–Unemotional						
Age Mean	95	−0.009	−2.756	**0.006**	0.084	740.735 ***
Age *SD*	94	−0.003	−0.194	0.846	0.002	751.111 ***
Scores Mean	64	0.005	0.588	0.556	0.002	572.123 ***
Scores *SD*	61	0.040	1.899	0.058	0.048	550.730 ***
Male % of sample	98	0.001	1.563	0.118	0.021	891.213 ***
Sample Size	104	0.000	−1.440	0.150	0.030	963.491 ***
Narcissism						
Age Mean	73	−0.009	−4.336	**0.000**	0.237	672.882 ***
Age *SD*	73	0.015	1.632	0.103	0.034	1172.357 ***
Scores Mean	52	−0.006	−0.942	0.346	0.024	643.070 ***
Scores *SD*	52	0.040	2.968	**0.003**	0.159	609.404 ***
Male % of sample	77	0.000	−0.139	0.889	0.000	1269.825 ***
Sample Size	82	0.000	0.677	0.498	0.003	1258.529 ***
Impulsivity						
Age Mean	70	−0.013	−4.303	**0.000**	0.287	438.220 ***
Age *SD* of Age	70	0.024	2.013	**0.044**	0.057	733.898 ***
Scores Mean	51	−0.011	−1.575	0.115	0.062	478.451 ***
Scores *SD*	49	0.060	4.068	**0.000**	0.325	254.423 ***
Male % of sample	73	0.000	0.295	0.768	0.002	757.510 ***
Sample Size	78	0.000	1.078	0.281	0.015	834.698 ***

*Notes. k* = Number of coefficient alphas for each subgroup of the moderator variable; *b* = unstandardized regression coefficient; *t* = significance test of moderator regression coefficient; *p* = *p* value of significance test, significant results are in bold; *R*^2^ = Total that proportion of variance accounts for; *Q*_E_ = Statistic for test of residual heterogeneity. *** *p* < 0.001.

## Data Availability

The original contributions presented in this study are included in the article/Appendix A. Further inquiries can be directed to the corresponding author.

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
