# Peer review of "A Reliability Generalization Meta-Analysis of the Antisocial Process Screening Device"

_behavsci, 2025, doi:10.3390/bs15070860_

Round 1

Reviewer 1 Report

Comments and Suggestions for Authors

The manuscript is well written, the research questions are clear, the research design is relevant for research questions, and the findings contribute to the research literature. I have only minor grammatical/APA style recommendations/questions which are listed below:

-line 106, there is a space between "from" and a comma

-APA style requires that et al. is used for multiple authors even the first time. So in line 155 it should be Tsang et al.

-I believe the "n"'s in figure 1 should be italicized

-line 216, I think it should be "psychology" rather than "psychological"

-line 394 reads awkwardly and I think "the" should be eliminated

-line 429-430 reads awkwardly, and I recommend "due to their incapactity to self-monitor"

-line 435-436, a citation is needed for the assertion that detained samples also tend to have a relatively low level of education

-line 455, it should be Fung et al.

-the reference section does not include issue numbers for journals

Author Response

The manuscript is well written, the research questions are clear, the research design is relevant for research questions, and the findings contribute to the research literature. I have only minor grammatical/APA style recommendations/questions which are listed below:

Response: Thank you for your comments and interests. These points have been addressed respectively (please see the following responses).

-line 106, there is a space between "from" and a comma

Response: We appreciate your careful reading. The spacing error in line 106 has been corrected (see Page 3).

“Coefficient alphas of the APSD total score have been most homogenous for the teach-er-report version (APSD-TR; ranging from, .86 to .97; Barrutieta & Prieto-Ursúa, 2015; Eremsoy et al., 2011), followed by the parent-report version (APSD-PR; from .73 to .89; Eremsoy et al., 2011; Frick et al., 2003), and the other-report version (APSD-OR; .73 to .88, from psychiatrists or other caregivers; de Wied, et al., 2014; Murrie et al., 2004).”

-APA style requires that et al. is used for multiple authors even the first time. So in line 155 it should be Tsang et al.

Response: We appreciate this important style correction. The citation in line 155 has been updated to use 'et al.' (see Page 4).

“Additionally, some research used translated versions of the APSD that had limited evidence for its translation quality (e.g., pilot studies or studies that tested the translation such as whether items were understandable and interpretable; Tsang et al., 2017).” 

-I believe the "n"'s in figure 1 should be italicized

Response: We appreciate you noting this important detail. We have revised Figure 1 to ensure all sample size notations ('n') are properly italicized (see Page 5).

-line 216, I think it should be "psychology" rather than "psychological"

Response: We appreciate this correction. The text in line 216 has been updated to 'psychology' as suggested (see Page 6).

“The coding process was completed by four trained psychology students. To ensure the accuracy of the coded dataset, 20% of the observations and the corresponding in-formation was selected at random and independently double-checked by two of the authors.”

-line 394 reads awkwardly and I think "the" should be eliminated

Response: Thank you for this helpful suggestion. We have revised the phrasing in line 394 by removing 'the' as recommended, which improves the flow of the sentence. The updated text now reads: “The internal consistency coefficients as indexed by Cronbach’s alpha of the APSD fluc-tuated across different application contexts.” (see Page 11).

-line 429-430 reads awkwardly, and I recommend "due to their incapactity to self-monitor"

Response: We appreciate this helpful suggestion. The text in lines 429-430 has been revised as recommended to read: 'due to their incapacity to self-monitor,' which improves the clarity of this section (see Page 12).

“On the other hand, as the APSD items focus on adolescent behaviors, observers (e.g., parents, teachers) demonstrate more reliable assessment of overt behavioral symptoms, while adolescent self-reports show limitations in evaluating their own symptoms due to their incapacity to self-monitor (Frick et al., 2000; Poythress et al., 2006a).”

-line 435-436, a citation is needed for the assertion that detained samples also tend to have a relatively low level of education

Response: Thank you for highlighting the need for a citation. We have added the following supporting reference (see Page 12).

“Detained samples also tend to have a relatively low level of education (Lankester et al., 2025), which limits their comprehension abilities when it comes to the CU items.”

Lankester, M., Coles, C., Trotter, A., Scott, S., Downs, J., Dickson, H., & Wickersham, A. (2025). The association between academic achievement and subsequent youth offending: A systematic review and meta-analysis. Journal of Developmental and Life-Course Criminology, 10, 477–500 https://doi.org/10.1007/s40865-025-00266-9

-line 455, it should be Fung et al.

Response: We appreciate your careful attention to citation formatting. The reference in line 455 has been updated to 'Fung et al. (2010)' (see Page 12).

“Nevertheless, Fung et al. (2010) have noted an insignificant correlation between psychopathic traits and age within a large sample of East Asian schoolchildren.”

-the reference section does not include issue numbers for journals

Response: We appreciate your careful review of our reference formatting. We have systematically updated the reference section to include issue numbers for journal articles, following APA style requirements (see References section).

Reviewer 2 Report

Comments and Suggestions for Authors

The current study examines the reliability of the Antisocial Process Screening Device (APSD) to measure psychopathic personality traits among adolescents. To do so, the authors conducted a reliability generalization meta-analysis across 158 studies. The models revealed several important findings including important differences in reliability between administration formats with the self-report version of the APSD demonstrating the lowest level of internal consistency. While this study has many strengths, there are some aspects of the manuscript that require further development including the argument for need of the study and conclusions drawn by the authors.

The authors state at least twice that APSD is the most widely used self- and other-report assessment of psychopathic personality traits in children and adolescents. However, the only source that is used to back this assertion is the Johnstone and Cooke (2004) study which is now over 20 years old. The authors use this fact as their primary argument for need of the study with the fairly old study being the only one cited at this point, that argument is fairly weak. Do you have a more recent study that demonstrates this point? There has been a movement among researchers to not categorize adolescents as having “psychopathic personality traits” due to stigma within society and instead call these traits in children “callous-unemotional traits.” Of course, this has not been a uniform movement towards the callous-unemotional traits operationalization, but many researchers have pivoted to using the Inventory of Callous-Unemotional Traits (ICU). It very well may be that the APSD is still the most commonly used instrument, it just needs to be addressed that there is variability in the literature as to how these traits are measured in children and adolescents and which is the more common.

The finding that self-reports have the lowest internal consistency of the administrative methods is very interesting. As the authors note, the self-report version has often been favored as the self-report scale’s reliability and validity. I would have liked to have seen more time discussing the discrepancy between your findings and the understanding within the field. You state on page 12 that, “Self-reports have been assumed to outperform other versions only when subjects can and do accurately assess their symptoms and behaviors (van Riezen & Segal, 1988). However, youths may have difficulty identifying and reporting their personal feelings correctly (Clark & Watson, 1991) which may increase the measure error of the self-report versions of the ASPD.” If that were true, wouldn’t we expect to see a follow up moderation analysis that looks includes the interaction between administration type and age? I would expect that as they got older they would be able to more accurately assess their behaviors.   

If you were to run that follow up analysis, you could look at how developmental stages impact these assessment types. I suspect that as the respondents get older as well that teachers and parents would have worse understanding of their behavior to your point: “On the other hand, as the APSD items focus on adolescent behaviors, observers may accurately assess the ASPD symptoms in those being observed, while adolescent subjects may less be aware of their own symptoms due to their lack in their capacity to monitor their behavior appropriately (Frick et al., 2000; Poythress et al., 2006a)” (p. 12).

Minor: I think this may be a formatting issue, your Callous-Unemotional Traits column with k and alpha were bolded but not any of the other subscales in Table 2, 3, and 4.

Author Response

The current study examines the reliability of the Antisocial Process Screening Device (APSD) to measure psychopathic personality traits among adolescents. To do so, the authors conducted a reliability generalization meta-analysis across 158 studies. The models revealed several important findings including important differences in reliability between administration formats with the self-report version of the APSD demonstrating the lowest level of internal consistency. While this study has many strengths, there are some aspects of the manuscript that require further development including the argument for need of the study and conclusions drawn by the authors.

Response: We sincerely appreciate your thoughtful and constructive feedback on our manuscript. In this revision, we have carefully addressed your key concerns regarding the study's rationale and conclusions, as detailed in our point-by-point responses below.

The authors state at least twice that APSD is the most widely used self- and other-report assessment of psychopathic personality traits in children and adolescents. However, the only source that is used to back this assertion is the Johnstone and Cooke (2004) study which is now over 20 years old. The authors use this fact as their primary argument for need of the study with the fairly old study being the only one cited at this point, that argument is fairly weak. Do you have a more recent study that demonstrates this point? There has been a movement among researchers to not categorize adolescents as having “psychopathic personality traits” due to stigma within society and instead call these traits in children “callous-unemotional traits.” Of course, this has not been a uniform movement towards the callous-unemotional traits operationalization, but many researchers have pivoted to using the Inventory of Callous-Unemotional Traits (ICU). It very well may be that the APSD is still the most commonly used instrument, it just needs to be addressed that there is variability in the literature as to how these traits are measured in children and adolescents and which is the more common.

Response: Thank you for this comment. In our revision, we have incorporated recent evidence supporting the APSD as one of the most widely used tools for assessing psychopathic traits in children and adolescents. We also acknowledge the variability in measurement tools (e.g., APSD, YPI, ICU) while emphasizing that the APSD remains among the one of most commonly used instruments in the literature (see Page 2).

“Several instruments were designed for use in children and adolescents, such as the Antisocial Process Screening Device (APSD; Frick & Hare 2001), Youth Psychopathic Traits Inventory—Child Version (YPI-CV; van Baardewijk et al. 2008), and the Inventory of Callous-Unemotional Traits (ICU; Frick, 2003) which is specifically capturing affective dimension of psychopathy. Among these instruments, the APSD was modeled directly after the Psychopathy Checklist-Revised (PCL-R; Hare, 2003). To date, the APSD (Frick & Hare, 2001; Frick et al., 1994) is one of the most widely used to assess psychopathic traits in children and adolescents.”

“The APSD is one of the most widely multi-informant measures for assessing psychopathy and antisocial behaviors in children aged 6 to 13 (Johnstone & Cooke, 2004; Kotler & McMahon, 2010; Salekin, 2017).”

The finding that self-reports have the lowest internal consistency of the administrative methods is very interesting. As the authors note, the self-report version has often been favored as the self-report scale’s reliability and validity. I would have liked to have seen more time discussing the discrepancy between your findings and the understanding within the field. You state on page 12 that, “Self-reports have been assumed to outperform other versions only when subjects can and do accurately assess their symptoms and behaviors (van Riezen & Segal, 1988). However, youths may have difficulty identifying and reporting their personal feelings correctly (Clark & Watson, 1991) which may increase the measure error of the self-report versions of the ASPD.” If that were true, wouldn’t we expect to see a follow up moderation analysis that looks includes the interaction between administration type and age? I would expect that as they got older they would be able to more accurately assess their behaviors.   

If you were to run that follow up analysis, you could look at how developmental stages impact these assessment types. I suspect that as the respondents get older as well that teachers and parents would have worse understanding of their behavior to your point: “On the other hand, as the APSD items focus on adolescent behaviors, observers may accurately assess the ASPD symptoms in those being observed, while adolescent subjects may less be aware of their own symptoms due to their lack in their capacity to monitor their behavior appropriately (Frick et al., 2000; Poythress et al., 2006a)” (p. 12).

Response: Thank you  for your interest and we appreciate your insightful comment and suggestion. Regarding the effects of administration type and age on reliability estimates, we conducted explanatory models to identify moderators accounting for variance in coefficient alphas for the APSD total and subscales (see page 10). For example, the analysis of the APSD total included three predictors in the multiple regression models: (1) administration format, (2) mean age of participants, and (3) score standard deviations. However, only administration format emerged as a significant predictor in the final model, while mean age showed no statistically significant association with reliability estimates.

Also, we added more clarifications about these points (see Page 12).

“On the other hand, as the APSD items focus on adolescent behaviors, observers (e.g., parents, teachers) demonstrate more reliable assessment of overt behavioral symptoms, while adolescent self-reports show limitations in evaluating their own symptoms due to their incapacity to self-monitor (Frick et al., 2000; Poythress et al., 2006a). These findings underscore the critical role of informant discrepancies in child psychopathology assessment, highlighted by the Attribution Bias Context Model (ABC Model) (De Los Reyes & Kazdin, 2005; De Los Reyes et al., 2015). For example, older children and adolescents typically demonstrate greater self-awareness of their psychological symptoms compared to younger children, leading to improved concordance between self- and observer-reports. Moreover, the magnitude of discrepancy may vary by symptom domain: externalizing problems generally show higher inter-rater agreement due to their observable nature, whereas internalizing problems exhibit greater discrepancies as they rely more heavily on subjective experience. These patterns highlight the need for future research utilizing comprehensive multi-informant, longitudinal designs to elucidate the nature of observed discrepancies in psychopathology assessment.”

Minor: I think this may be a formatting issue, your Callous-Unemotional Traits column with k and alpha were bolded but not any of the other subscales in Table 2, 3, and 4.

Response: Thank you for your careful review. We have corrected the formatting inconsistency in Tables by ensuring uniform presentation of all subscale columns to maintain consistency throughout the manuscript.